# The Importance of *Drosophila melanogaster* Research to UnCover Cellular Pathways Underlying Parkinson’s Disease

**DOI:** 10.3390/cells10030579

**Published:** 2021-03-06

**Authors:** Melissa Vos, Christine Klein

**Affiliations:** Institute of Neurogenetics, University of Luebeck, Ratzeburger Allee 160, Building 67, 23562 Luebeck, Germany

**Keywords:** Parkinson’s disease, *Drosophila melanogaster*, mitochondria, endo-lysosomal pathway, lipid homeostasis

## Abstract

Parkinson’s disease (PD) is a complex neurodegenerative disorder that is currently incurable. As a consequence of an incomplete understanding of the etiology of the disease, therapeutic strategies mainly focus on symptomatic treatment. Even though the majority of PD cases remain idiopathic (~90%), several genes have been identified to be causative for PD, facilitating the generation of animal models that are a good alternative to study disease pathways and to increase our understanding of the underlying mechanisms of PD. *Drosophila melanogaster* has proven to be an excellent model in these studies. In this review, we will discuss the different PD models in flies and key findings identified in flies in different affected pathways in PD. Several molecular changes have been identified, of which mitochondrial dysfunction and a defective endo-lysosomal pathway emerge to be the most relevant for PD pathogenesis. Studies in flies have significantly contributed to our knowledge of how disease genes affect and interact in these pathways enabling a better understanding of the disease etiology and providing possible therapeutic targets for the treatment of PD, some of which have already resulted in clinical trials.

## 1. Introduction

Parkinson’s disease (PD) is the second most common neurodegenerative disorder after Alzheimer’s disease and is characterized by bradykinesia, tremor and rigidity [1,2]. These motor signs arise from a lack of dopamine due to a decline of dopaminergic neurons in the substantia nigra [3]. Furthermore, the majority of PD patients present with Lewy bodies, the pathological hallmark of PD that are aggregates of proteins–mostly consisting of alpha-synuclein-, but also containing mitochondria and lipids [4]. In addition to motor symptoms, a wide range of non-motor symptoms occur, including dementia, depression, constipation and sleep problems [5,6,7,8]. These non-motor symptoms may occur years prior to the initial diagnosis [7] and strongly impact the quality of life [9].

PD is a progressive disorder, the prevalence of which increases with age and globally affects 1% of the population above 60 years [10,11,12] imposing a heavy burden on patients, caregivers, and society. An exponential increase in PD prevalence can be expected, in part due to an increasingly aging society [13]. Hence, therapeutic strategies that halt or reverse the disease are invaluable. In the past decades, major advances have been made in our understanding of the disease pathogenesis. However, unfortunately, important gaps remain to be further elucidated and, thus, alleviating signs and symptoms via e.g., dopamine replacement therapy is currently the major therapeutic strategy [14]. To increase our understanding of the underlying pathways resulting in PD, animal models that present with the basic PD-like features, including dopaminergic neuron loss and motor signs are key. Studies in these animal models provide novel insights into disease mechanisms and identify novel therapeutic targets. Unfortunately, an animal that completely represents all signs and symptoms does not exist. Animal models created in different species often reproduce one or several PD-related signs and symptoms, while lacking other essential features. A common and remarkable feature in different genetic mouse models of PD is the lack of dopaminergic neurodegeneration [15,16,17,18], suggestive of a protective mechanism in mice that is absent in humans and pointing to the fact that each animal model has shortcomings when compared to model human diseases.

Nonetheless, *Drosophila melanogaster* is an excellent model to study the molecular mechanisms in PD. The fruit fly is a relatively simple animal; however, it does contain a complex neuronal circuitry including clusters of dopaminergic neurons [19,20]. *D. melanogaster* undergoes different stages (egg, larva, pupa, fly) in a short lifecycle of around 10 days (Table 1). Furthermore, many experimental tools exist to genetically manipulate *D. melanogaster* allowing a relatively rapid creation of genetically modified flies that can serve as a disease model. In this review, we focus on the advantages of using *D. melanogaster* as animal model to study disease pathways and the key findings in the molecular mechanisms in PD that arose from studies on the fruit fly. First, we will discuss the possibilities of using *Drosophila* in the study of a variety of human diseases, followed by an overview of the different available PD fly models. Next, we will review in more detail the identification of altered molecular mechanisms in PD fly models with a focus on mitochondrial dysfunction, defective endo-lysosomal pathway and lipids that may constitute a linking factor between these two mechanisms. Finally, the relevance of these findings for PD patients and its consequence for therapeutic strategies will be discussed.

## 2. *Drosophila melanogaster* as Animal Model to Study Human Diseases

Fruit flies are small animals, which facilitates live imaging enabling in vivo recording of biological processes that can further clarify disease-relevant mechanisms, including analyzing dopaminergic neurons that are mainly affected in PD (Table 1). The genome of flies consists of only four chromosomes, which is a markedly lower number than the 23 chromosomes in the human genome. Nonetheless, more than 75% of the human disease-causing genes have an ortholog in flies (Table 1) [21]. Using established methodology to manipulate genes, including the Crispr cas9 system that allows the introduction of specific point mutations [22,23] or the yeast UAS-gal4 system enabling overexpression or knockdown of proteins ubiquitously or in relevant tissue (Table 1) [24,25,26], the proper genetic allele can be created that mimics the genetic disease context to study essential biological processes. For example, overexpression of a specific protein may be more relevant for gain-of-function mutations than knocking down that specific protein. In addition, several collections of mutant flies exist or can be created via toxin-induced mutagenesis. These are especially useful to perform modifier screens, which is one of the benefits of using *D. melanogaster* in the study of diseases. These modifier screens enable the identification of genes that interact with disease-related mechanisms contributing to the elucidation of novel pathways involved in or providing promising therapeutic targets. Such modifier screens have proven their value in PD research with the identification of vitamin K2, aconitase, sterol regulatory element-binding protein 1 (SREBP1) (see below) [27,28,29]. Furthermore, screens in which specific phenotypes are analyzed that mimic phenotypes of a specific disease model can lead to the identification of new genes that are involved in the disease pathogenesis (Figure 1).

The creation of new fly models often emerges from the identification of a disease-causing gene. For example, loss-of-function mutations in *Thousand and one amino acid kinase 1* (*TOAK1*) have been identified to be causative for neurodevelopmental delay and intellectual disability [30]. Knockdown of the *Drosophila* ortholog Tao shows a delayed development and lethality at the final larval stage accompanying defective brain and neuronal morphology [30] mimicking the disease phenotype. Furthermore, knockdown of Tao presents with altered mitochondrial distribution in axons of motor neurons [30], suggesting mitochondria play a role in the disease pathogenesis, which was confirmed in patient-derived fibroblasts that display defective mitochondrial morphology [30]. Thus, this fly model for neurodevelopmental delay can be used to further investigate the relationship between abnormal mitochondria and brain morphology that results in neurodevelopmental delay. Notably, *D. melanogaster* can also be used to model diseases that do not occur naturally in flies. Lethal tumor growth and metastasis have not been observed in wild type flies [21]. Nonetheless, genes that play a role in tumor formation in humans, such as those affecting the cell-cycle control have been identified in flies and are being used to study processes of tumor growth [21]. These examples show that *D. melanogaster* can be employed to study a plethora of human diseases even if the exact disease process does not occur in this animal.

## 3. *Drosophila melanogaster* as Animal Model for Parkinson’s Disease

Many animal models exist to study PD. Non-human primate models are the most closely linked to humans and, thus, findings in primate PD models will most likely be resembling those present in PD patients the most. Compared to rodent models, flies are a simple animal model, however, it does represent many of the key features of PD, in contrast to rodent models that often lack loss of dopaminergic neurons [15,16,17]. For a long time, PD was considered to be a sporadic disorder; however, with the identification of *SNCA* to be causative for PD in the late 90s [31], the importance of genes in the etiology of PD became obvious and has been constantly growing in the past 25 years. Since then, various genes have been identified to be causative for PD (Table 2) [32] and have enabled the creation of genetic animal models of PD leading to a significant increase in our knowledge of its etiology. In addition, many genes have been linked previously to PD; however, were not confirmed or were disproven to be causative for PD [32,33]. Nonetheless, several of these proteins interact with or function in shared mechanisms with the PD-confirmed proteins and thus can provide more information about the affected pathways relevant for PD and, hence, will also be discussed in the relevant sections below.

Several fly models to study the underlying mechanisms of PD exist to mimic the human condition, including genetic models (overexpression, knockdown and endogenous) and toxin-induced models. A well-established overexpression model is the overexpression of human alpha-synuclein. Interestingly, *SNCA* is the only PD gene that does not have a fly ortholog. Nonetheless, the fly models that have been created overexpress wild type-and disease-mutant human *SNCA* using the UAS-Gal4 system [34] and develop PD-like symptoms, suggesting that the underlying disease mechanisms are evolutionarily conserved. Furthermore, mutations in *SNCA* causing PD are gain-of-function mutations [35], making the overexpression model a relevant one. In addition to these causative genes, risk factors have been identified that increase the susceptibility for PD (Table 2). While risk factors are not disease-causing, studies that investigate the functional pathways of these genes are useful to identify mechanisms that are linked to PD. The best-studied risk factor for PD is glucocerebrosidase (GCase) encoded by *GBA* [36,37]. Remarkably, flies have two *GBA* orthologs (*gba1a* and *gba1b*) [38]. To study the function of GCase and how it increases the risk for PD, loss-of-function alleles of both fly genes are being used [38,39] as well as overexpression of human GCase-carrying mutations that are known to increase the risk for PD [40]. This demonstrates that studying the effect of different gene dosages is valuable in the investigation of protein function or involved pathways. Many genes and risk factors are being used in studies to dissect the PD pathogenesis; however, the vast majority of PD cases are currently considered idiopathic. Nonetheless, studies using genetic fly models are useful for various reasons in this context, as well: 1. PD-causing genes have been shown to be altered in sporadic PD [20,41,42,43], suggesting they are functioning as risk factors in these forms of PD; 2. Idiopathic forms of PD show a similar disease progress compared to some of the genetic forms of the disease [44,45], suggesting similar underlying mechanisms to play a role; 3. Mitochondrial defects have first been identified in a subset of idiopathic patients [46] and were later confirmed in genetic forms of PD [47,48] and in animal models [49,50,51,52] to play a role in at least a subset of PD patients. Many of the existing PD fly models reflect PD-related signs and symptoms and their relevance for the disease has been validated numerously. For example, overexpression of human wild type PINK1 in *pink1*-mutant flies is able to rescue the phenotypes, while overexpression of human mutant PINK1 is not [52,53], suggesting that the function of PINK1 is evolutionarily conserved and that flies are a valid animal model to investigate the function of PINK1 in relation to PD.

Mitochondrial toxins, like 1-methyl-4-phenyl-1,2,3,6-tetrahydropyridine (MPTP) [54,55] or pesticides such as rotenone [56,57,58] induce parkinsonian-like symptoms that resemble idiopathic forms of PD and can be used to model idiopathic PD in flies [59]. MPTP specifically affects dopaminergic neurons [60], while rotenone is a systemic mitochondrial drug without any preference towards a specific type of neuron or organ that also results in the loss of dopaminergic neurons [59,61]. 6-hydroxydopamine (6-OHDA) is a drug that selectively destroys dopaminergic neurons and is often used to model PD. 6-OHDA cannot pass the blood-brain barrier and therefore, has to be injected into the brain [62]. This mode of application of the drug makes it an unfavorable drug to apply in the fruit fly. Nonetheless, it needs to be considered that application of these drugs to the fly medium is often done in an acute, high-dose fashion, which possibly activates additional mechanisms. In contrast, patients are typically chronically exposed to these drugs for years at a low dose before symptoms arise.

## 4. The Mitochondrial-PD Connection in *Drosophila melanogaster*

The first observation that mitochondria are involved in PD came from the observation in the 80s that drug abusers using drugs contaminated with MPTP developed parkinsonian-like symptoms [63]. MPTP and rotenone specifically inhibit complex I of the electron transport chain (ETC) resulting in disrupted energy production [64,65,66]. Later, defective complex I activity was observed in post mortem brains of idiopathic PD patients [67] and PD-causing genes were identified that are linked to mitochondrial function, including *PINK1* [47], *Parkin* [68] and *DJ-1* [48] that cause autosomal recessive PD (Table 2). Here, we will discuss the advances that studies using *pink1*- and *parkin*-mutant flies have contributed to our knowledge of their shared functions, including mitochondrial remodeling and mitophagy, as well as their independent functions, and the role of DJ-1 in oxidative stress (Figure 2).

### 4.1. The Pink1-Parkin Pathway

Patients suffering from PINK1 and Parkin-related PD both present with early-onset forms of PD, show a similar disease progression [2] and respond comparably well to therapy. In addition, flies with loss of Pink1 and Parkin function share phenotypic similarities, including locomotion defects, mitochondrial dysfunction and abnormal mitochondrial morphology [49,50]. These observations support the notion that these genes function in the same pathway. Overexpression of Parkin in *pink1*-mutant flies rescued the pathological phenotypes in the Pink1-deficient flies, whereas overexpression of Pink1 did not induce a rescue in *parkin*-mutant phenotypes [49,50]. These studies in flies were the first to provide experimental evidence that PINK1 and Parkin function in the same pathway with Parkin acting downstream of PINK1 to maintain mitochondrial integrity. These findings were later confirmed in cellular models (see Section 4.1.2) [69,70].

#### 4.1.1. Mitochondrial Fusion and Fission

The swollen, enlarged mitochondria hint toward a function in mitochondrial remodeling that consists of regulated fusion and fission of mitochondria. Several evolutionarily conserved genes that express GTPase proteins have been identified to be involved in these fusion and fission events. Mitofusins 1 (Mfn1) and 2 (Mfn2) are located at the outer mitochondrial membrane and promote fusion at the outer mitochondrial membrane, optic atrophy 1 (OPA1), located at the intermembrane space provokes fusion of the inner mitochondrial membranes [71,72]. Dynamin-related protein 1 (Drp1) is a cytosolic GTPase that is important for mitochondrial fission [71,73]. Studies that apply genetic alterations of these genes in *pink1*- or *parkin*-mutant flies have confirmed the contribution of the Pink1-Parkin pathway in mitochondrial morphology, i.e., stimulation of fission or inhibition of fusion improve the phenotypes, while promotion of fusion or preventing fission events results in the lethality of Pink1- and Parkin-deficient flies [74,75]. Remarkably, recent findings in flies show that PINK1-dependent phosphorylation of Drp1 regulates mitochondrial dynamics that are independent of Parkin [76]. While these data show that loss of Pink1 and Parkin impede mitochondrial remodeling, loss of Drp1 results in distinct mitochondrial phenotypes [77] compared to loss of Pink1 or Parkin [49,50,74,75,78], suggesting that under healthy conditions, Pink1 and Parkin are not part of the key factors of drp1-mediated fission, but are rather activated under pathological conditions.

#### 4.1.2. Mitophagy

The mitochondrial fusion-fission machinery is an important tool in the facilitation of clearing damaged mitochondria [79,80,81], as it selects out defective mitochondria and reduces the mitochondrial size to enable mitochondrial degradation, a process termed mitophagy. Interestingly, Parkin, a ubiquitin ligase [82], ubiquitinates target proteins (that may be located on various organelles such as mitochondria) to serve as a signal for these ubiquitinated proteins and organelles to be degraded [83,84]. Follow-up studies in cellular models have provided further clarification of PINK1-Parkin-related mitophagy. Under healthy conditions, PINK1 is immediately cleaved and imported into mitochondria via the TIM-TOM complex. In defective mitochondria that lead to depolarization of mitochondria, PINK1 remains uncleaved at the outer mitochondrial membrane followed by (auto)phosphorylation of PINK1, a kinase, and Parkin, providing a signal for Parkin to translocate to the mitochondria resulting in mitochondrial degradation [69,85]. These findings were also confirmed in a *Drosophila* cellular model [86]. Furthermore, studies in these cellular fly models provided evidence for the ubiquitination of Mfn by Parkin to be the critical labeling step for mitochondrial degradation [85,86,87]. F-box domain only protein 7 (Fbxo7) functions in mitophagy via direct interaction with Pink1 and Parkin [88]. Wild type Fbxo7 facilitates mitophagy, while mutant Fbxo7 impedes mitophagy [89]. Mutations in *FBXO7* cause atypical parkinsonism and flies do not have a Fbxo7 protein, but overexpression of the human wild type protein rescues Parkin-deficient phenotypes in flies [88], further supporting the role of Fbxo7 in mitophagy. Remarkably, overexpression of wild type Fbxo7 in flies induces age-dependent locomotion defects, dopaminergic neuron loss and swollen mitochondrial [89], mimicking PD-like symptoms.

For a long time and due to a lack of appropriate tools, it was only possible to study mitophagy events in a cellular system that required the application of mitochondrial drugs to induce mitophagy. The main drawback of this experimental setup was that this did not represent physiological conditions in the disease context. Following the creation of mt-Keima, a pH-sensitive sensor that is targeted to mitochondria [90], flies were created that express mt-Keima creating an in vivo model to study mitophagy [91,92]. Interestingly, studies revealed that loss of Pink1 or Parkin did not display an effect on mitophagy levels in the initial life stages of flies [91,92]; however, basal mitophagy levels were reduced in aging flies [91]. Hence, these data are suggestive of compensatory mechanisms maintaining the basic levels of mitophagy initially. Nonetheless, this compensatory mechanism cannot fulfill the requirements for mitophagy resulting in the age-dependent lowering of mitophagy levels following loss of PINK1 or Parkin.

The relevance of mitophagy in disease pathogenesis remains to be further clarified. Overexpression of Parkin in *pink1*-mutant flies rescues pathological phenotypes [49,50], independent of the Pink1-dependent translocation of Parkin to the mitochondria, which is absent in Pink1-deficient flies. These observations cannot easily be reconciled with an exclusive function of Pink1 and Parkin in mitophagy as the single underlying disease mechanisms. Furthermore, loss of PINK1 or Parkin results in defects in mitophagy [91] and both PINK1 and Parkin are involved in functions that are independent of one another (see Section 4.1.3 and Section 4.2). In contrast, wild type Fbxo7 that stimulates mitophagy, induces age-dependent PD-like symptoms in flies, challenging the theory that ablation of mitophagy plays a major role in PD pathogenesis. Studies in flies expressing mutant Fbxo7 would aid to clarify this discrepancy. Collectively, these data are indicative of yet to be discovered additional mechanisms to play a role in disease pathogenesis.

#### 4.1.3. Mitochondrial Complex I

Loss of Pink1 results in lower complex I activity [52] that is found to be absent in Parkin-deficient flies [93]. Interestingly, the stimulation of mitochondrial fission that is enabling mitophagy [79,80,81], does not induce a rescue of complex I activity in *pink1*-mutant flies [93,94]. Bypassing the defective complex I in Pink1-deficient flies via the expression of the yeast complex I, ndi1, rescues the mitochondrial defects in these flies but not in *parkin*-mutant flies [93]. These data provide evidence that defective complex I plays an important role in PINK1-related PD pathogenesis. Moreover, phosphorylation of ND-42 (the fly ortholog of NdufA10), a complex I subunit that is necessary for proper complex I function, is Pink1-dependent [95], directly linking Pink1 to complex I. These data were confirmed and further validated in patient-derived fibroblasts carrying *PINK1* mutations [95], suggesting that the mitochondrial ETC serves as a therapeutic target in PINK1-related PD. Also here, studies in flies have contributed significantly. Application of near-infrared light that boosts ETC complex IV improves the mitochondrial phenotypes in *pink1*-mutant flies [96] and studies in a mammalian alpha-synuclein model confirmed that this photostimulation is protective for dopaminergic neuron loss [97]. Further studies need to be performed to test the effectiveness of this therapy in PD patients. However, the downside of this therapeutic strategy is the invasiveness to apply the infrared light. Hence, drugs that stimulate the ETC would be the more preferred strategy. Studies in flies identified vitamin k2 as an alternative electron carrier molecule that similar to ubiquinone can stimulate the ETC resulting in a rescue of Pink1-related phenotypes [27], nominating it to be a promising therapeutic strategy in the treatment of PINK1-related PD patients. The application of ubiquinone and vitamin k2 as therapy in patients has been initiated in clinical trials [98,99] and serves as an excellent example of therapeutic targets identified in flies that are now being tested in patients.

One of the first observations in post mortem brains of PD patients was the accumulation of iron in the substantia nigra [100]. However, how iron-mediated toxicity contributes to the PD pathogenesis remains enigmatic. In a genetic modifier screen, mitochondrial aconitase (acon) was identified to improve the phenotypes in *pink1*-mutant flies [28]. Acon functions in the Krebs cycle [101] and harbors iron-sulfur clusters that are inactivated in loss of Pink1 resulting in the release of iron and reactive oxygen species in the form of hydroxyl peroxide to induce mitochondrial dysfunction [28]. More recent studies showed that increasing the bioavailability of these iron-sulfur clusters in the mitochondrial matrix promotes mitochondrial respiration [102], directly linking iron, oxidative stress and mitochondrial respiration.

### 4.2. Parkin-Specific Functions

E3 ubiquitin ligases, like Parkin, interact with specific ubiquitin activating (E1) and ubiquitin-conjugating (E2) enzymes to label proteins or organelles for the ubiquitination degrading pathway [103]. Studies in flies and mammalian cells have revealed Rad6, a protein involved in mental retardation, to be one of the E2 conjugating enzymes that bind to Parkin. Lack of Rad6 prevents the translocation of Parkin to the mitochondria and therefore the ubiquitination of mitochondrial proteins [104]. Deubiquitinases oppose that function by removing the ubiquitin from ubiquitin-labeled proteins [105]. USP15, a deubiquitinase specifically counteracts the parkin-mediated ubiquitination of mitochondria and thus prevents mitophagy, while USP15 knockdown stimulates mitochondrial ubiquitination and age-dependent mitophagy defects in Parkin-deficient muscles and dopaminergic neurons [91,106]. In addition, Mask, an anchoring protein also known as ANKHD1 (Ankyrin repeats and KH domain containing protein 1) has been identified in flies to genetically interact with Parkin to inhibit mitophagy [107]. One of the mitochondrial ubiquitination substrates for Parkin is voltage-dependent anion-selective channel 1 (VDAC1) [70,108] and polyubiquitination of VDAC1 [109] triggers mitophagy via the recruitment of p62 and LC3 to the mitochondria [108]. VDAC1 is a component of the mitochondrial transition pore and controls the opening of this pore in the regulation of apoptosis [110]. Interestingly, the first phenotypic characterization of Parkin-deficient flies shows apoptotic cell death in flight muscle [78], providing experimental evidence for Parkin to function in apoptosis. Furthermore, monoubiquitination of VDAC1 is important in the regulation of apoptosis [109] and lack of monoubiquitination of VDAC1 phenocopies parkinsonism, including locomotion defects and loss of dopaminergic neurons [109], suggesting that regulation of apoptosis plays a role in disease pathogenesis. Parkin is a cytosolic protein and thus non-mitochondrial ubiquitination is to be expected. Research in mammalian *Parkin*-mutant models shows Parkin-dependent PARIS (ZNF746) ubiquitination to regulate mitochondrial biogenesis via peroxisome proliferator-activated receptor gamma coactivator 1-alpha (PGC-1a) [111]. Follow-up studies in flies elaborate on this role in mitochondrial biogenesis and reveal that both Pink1 and Parkin promote mitochondrial health via the PARIS-dependent PGC-1a regulatory function on mitochondrial biogenesis [112], suggesting regulators of mitochondrial biogenesis can lead to new therapeutic avenues.

### 4.3. The Role of Oxidative Stress in PD

DJ-1 is a mitochondrial-linked protein and flies carry two *DJ-1* genes, *dj-1α* (predominantly expressed in testes) and *dj-1β* (ubiquitously expressed) (Table 2). Homozygous loss of dj-1β results in motor defects that deteriorate with oxidative stress [113,114,115] confirming its function in controlling intracellular reactive oxygen species [116,117]. Double knock-out flies for *dj-1α* and *dj-1β* results in a variety of mitochondrial phenotypes that phenocopy those observed in Pink1- and Parkin-deficient flies [49,50,78,118], suggesting a genetic link between these three proteins. Upregulation of DJ-1 improves *pink1*-mutant phenotypes, however does not rescue *parkin*-mutant flies [118], suggesting a specific interaction between PINK1 and DJ-1, independent of Parkin. RNA sequencing analyses in *dj-1β*-mutant flies show decreased levels of NADP+ -dependent isocitrate dehydrogenase (IDH) inducing hypersensitivity to oxidative stress followed by age-dependent mitochondrial defects [119]. IDH uses the reducing potency of NADPH to suppress intracellular and mitochondrial ROS responsible for the mitochondrial phenotypes [119]. The ablation of reducing power is a feature common to that observed upon loss of Pink1 with Acon (see Section 4.1.3) [28] providing more experimental support for the importance of oxidative stress in PD.

## 5. Endo-Lysosomal PD Connection in *Drosophila melanogaster*

The implication of mitochondrial dysfunction was the earliest mechanistic insight into PD pathogenesis, however, an increasing amount of evidence is indicative of an important role of the endo-lysosomal pathway in PD [120,121,122]. The endo-lysosomal pathway is important for protein homeostasis and consists of endocytosis, retromer and autolysosome [123,124,125] that we will separately discuss. PD-related proteins that are linked to one or more of these processes are alpha-synuclein, VPS35 and LRRK2 that cause autosomal dominant forms of PD (Figure 3). Interestingly, low-penetrance variants of alpha-synuclein and LRRK2 have been found in idiopathic forms of PD, supporting the importance of this pathway in PD.

### 5.1. Vesicle Trafficking Endocytosis

Using an alpha-synuclein yeast PD model, defects in the trafficking between the endoplasmic reticulum (ER) and Golgi were observed [126]. *SNCA* encodes alpha-synuclein that contains a mitochondrial targeting signal [127]. In line with this, alpha-synuclein models have shown oxidative and mitochondrial stress [127], providing overlapping features with other forms of PD that are clearly linked to mitochondrial dysfunction. However, mitochondrial dysfunction appears to be a secondary effect rather than the initial defect related to *SNCA* mutations. Alpha-synuclein models were the first genetic PD model in flies [34], mimicking the typical PD-like symptoms, including locomotor deficits, dopaminergic neuron loss and alpha-synuclein-aggregated inclusion bodies [34,128,129]. Furthermore, expression of alpha-synuclein in flies results in Ser129 phosphorylated alpha-synuclein [130], similar to what was previously observed in PD patients [131,132]. Phosphorylation increases with aging flies [130] and is crucial for alpha-synuclein-induced dopaminergic neuron loss and inclusion formation that exhibit a neuroprotective role [133].

Combinational studies using yeast and *Drosophila* alpha-synuclein models for the first time linked Rab proteins to alpha-synuclein [134]. Rab proteins are a family of GTPases that are a member of the Ras superfamily of small G proteins [135]. Rab proteins function in membrane trafficking and more than 60 different Rab proteins are identified that differ according to their subcellular localization [135]. Rab1, the endosomal recycling factor Rab11, the late endosomal Rab7 and post Golgi vesicle trafficking Rab8 are neuroprotective or improve locomotion in flies expressing alpha-synuclein [134,136,137,138] providing ample experimental evidence for a function of alpha-synuclein in vesicle trafficking. Another example of a PD-related protein that is linked to Rab proteins is LRRK2 [139]. LRRK2 is a kinase that affects endocytosis via the LRRK2-dependent phosphorylation of Endophilin A [140,141], an endocytotic protein [142] and Synaptojanin [143] that plays an essential role in synaptic vesicle recycling [144] further supporting vesicle trafficking to be a common function in PD-related proteins. Flies carry a single *Leucine Rich Repeat Kinase 2* (*LRRK2*) ortholog, *LRRK* [128]. Several models have been created in flies to model LRRK2-related PD. One is the knock-out of *LRRK* that allows the study of the physiological function of LRRK and shows that LRRK is localized at endosomal and lysosomal membranes [139]. In flies, studies report contradictory results in relation to dopaminergic neurons and locomotion in loss of LRRK [145,146,147]. However, the expression of PD-related mutant *Drosophila* LRRK or human LRRK2 induces dopaminergic neuron loss and its related locomotion defects [128]. Remarkably, mutations in Synaptojanin induce loss of dopaminergic neurons in flies but do not induce defects in synaptic vesicle endocytosis [143] mystifying the role of vesicle trafficking in PD pathogenesis. Nonetheless, loss of LRRK induces enlarged endosomes and accumulation of autophagosomes that can be rescued by the stimulation of retromer-dependent recycling from late endosomes to the Golgi apparatus [146]. Furthermore, the PD-related protein VPS35 interacts with LRRK2 in the Rab-mediated endocytic pathway [148], linking these PD genes and providing evidence for the importance of the Rab-dependent regulation of retromer-mediated endocytosis in the pathogenesis of PD.

### 5.2. The Retromer Complex

The retromer is a highly conserved, multi-protein complex that is required for the recycling of transmembrane receptors from endosomes to the trans-Golgi network [149,150,151]. Its recruitment to the endosomal membrane is carried out by the sorting nexin dimer component of the complex [152]. The cargo recognition and binding function are executed by the core vacuolar protein sorting trimer, VPS35, VPS29 and VPS26 [153]. Interestingly, Rab7 binds to this trimer facilitating the recruitment of the retromer to endosomes in concert with the sorting nexin dimer [154], stipulating a possible explanation in which LRRK2 and VPS35 are functionally linked with each other.

One of these core proteins, VPS35 causes autosomal dominant PD [153] and null mutants in flies are lethal [148] underlining the importance of VPS35. Studies that investigate the effects of expression of PD-related mutant *VPS35* are conflicting. In one study, these flies induce age-dependent loss of dopaminergic neurons and locomotion deficits [155], while another did not observe such parkinsonian-like phenotypes [156]. However, the latter group found that mutant *VPS35* functions in a dominant-negative fashion, which is supported by multiple studies in fly or mouse systems [157,158,159,160]. Furthermore, knockdown of VPS35 aggravated locomotor deficits and provoked accumulation of insoluble alpha-synuclein due to defective lysosomal degradation in alpha-synuclein-expressing flies [161], while overexpression of VPS35 rescues locomotion in LRRK2- and Parkin-models [156,159].

iPLA2-VIA, the *Drosophila* ortholog of PLA2G6 interacts with the retromer by binding to VPS35 and VPS26 to induce protein and lipid recycling [162]. Mutations in the phospholipase gene *PLA2G6* cause different neurodegenerative disorders that, next to shared defects, also result in distinct symptoms [163]. Flies deficient of PLA2G6 show defective locomotion, (dopaminergic) neurodegeneration and defective synaptic transmission [162,164,165,166]. Furthermore, mitochondrial dysfunction and membrane abnormalities are induced by loss of iPLA1-VIA [164] that are most likely secondary phenotypes following upstream defects that are related to its function as a phospholipase that hydrolyzes glycerol phospholipids. iPLA2-VIA deficiency in fly brains does not affect the phospholipid composition [162], but results in phospholipids with shorter acyl chain length [165]. The shortened acyl chain causes ER stress and alpha-synuclein aggregation that can be rescued by restoring the brain lipid composition [165]. Furthermore, loss of iPLA-VIA increases ceramide levels and lowering these ceramide levels suppresses neurodegeneration [162]. Ceramide is the basic sphingolipid synthesized in the ER. Following its transport to the Golgi, ceramide is converted to ceramide phosphoethanolamine (CPE) or sphingomyelin (SM) [167]. CPE/SM are recycled via the retromer complex [168,169] that requires binding of VPS35 to the retromer [162]. CPE/SM that is not undergoing the recycling via the retromer will be relocalized to the lysosomes for hydrolytic breakdown to ceramides and sphingolipids [170]. Impaired retromer function, via loss of iPLA2-VIA or PLA2G6, thus mitigates the CPE/SM recycling, inducing increased lysosomal ceramide levels [122,162].

Retromer depletion impairs the maturation of cathepsin D (CTSD), a lysosomal protease that functions in the metabolism of alpha-synuclein leading to lysosomal accumulation of alpha-synuclein [161]. Knockdown of VPS35 aggravates the locomotion deficits in alpha-synuclein-expressing flies [161]. Interestingly, human cells expressing alpha-synuclein show reduced VPS26 and VPS35 protein levels and increased lysosomal stress that is similar to those observed in PLA2G6 deficiency [162] suggesting alpha-synuclein plays a role in sphingolipid metabolism [122]. Mammalian alpha-synuclein cells indeed show increased ceramide levels and myriocin, to inhibit ceramide synthesis, improves alpha-synuclein-induced degeneration in flies [162]. These mechanistic studies link LRRK2, VPS35, PLA2G6 and alpha-synuclein to the retromer providing strong evidence for the retromer, an important regulator of sphingolipid metabolism [170], and therefore also of sphingolipid ceramide, to play a central role in PD pathogenesis. Interestingly, ceramide is an important regulator of autophagy [171].

### 5.3. The Autolysosome

Autophagy is tightly regulated. Protein aggregates or damaged organelles are targeted and packed in the autophagosome followed by the fusion of the autophagosome with lysosomes to form the autolysosome that enables the degradation of protein aggregates or damaged organelles [172]. The LRRK2-dependent phosphorylation of Endophilin A recruits Atg3 (autophagy-related 3) to promote autophagy at the synapse [140,141]. Furthermore, Synaptojanin is a phosphatase that dephosphorylates PI(3) and PI(3,5)P2, two lipids localized on autophagosomal membranes. *Synaptojanin*-mutant flies show defective dephosphorylation [143], indicating autophagosomal abnormalities and suggesting an important role for defective autophagy in LRRK2 PD. Autophagy-related or lysosomal proteins are important for the activation, flux and completion of autophagy, including the transmembrane lysosomal ATPase ATP13A2 [172,173,174]. Mutations in *ATP13A2* lead to juvenile-onset PD, or Kufor-Rakeb syndrome [175]. A combinational study using mice and flies that are deficient for ATP13A2 show that ATP13A2 is involved in the proper fusion of the autophagosome with the lysosome in a histone deacetylase 6 (HDAC6)-dependent manner [176]. This fusion is an important step for the degradation of proteins or organelles in autophagy [177]. Interestingly, HDAC6 is protective for alpha-synuclein-induced dopaminergic degeneration [178].

## 6. Lipids as Connecting Factor between Mitochondrial Dysfunction and a Defective Endo-Lysosomal Pathway in *Drosophila melanogaster*

As discussed above, research revealed two seemingly distinct molecular pathways in PD. However, how these are connected to converge to the same signs and symptoms remains mostly enigmatic. Recently, emerging evidence highlight the importance of lipids in PD. Lipids are important for proper mitochondrial function and are key in the endo-lysosomal pathway and may constitute the missing link between mitochondria and endo-lysosomes in relation to PD. Numerous studies report lipid alterations in PD patients [4,179] and PD-related models [162,180,181]. A genome-wide association study identified sterol regulatory element-binding protein 1 (SREBP-1) as a risk locus for PD [182]. Furthermore, a screen to identify genes that promote mitophagy in *Drosophila* S2 cells revealed SREBP-1 to regulate mitophagy [29]. SREBP-1 is a regulator of lipogenesis and these data support distorted regulation of mitophagy to employ a risk factor for the development of PD. Furthermore, a screen to identify modifiers of Pink1, showed that partial genetic or pharmacological inhibition of fatty acid synthase 2 (FASN2) improves Pink1-deficient phenotypes that are evolutionarily conserved [181]. Lower FASN2 is indirectly linked to increased cardiolipin levels [181]. Cardiolipin is a mitochondrial-specific lipid residing at the inner mitochondrial membrane [183,184,185] that is decreased in *pink1*-mutant flies and supplementation of cardiolipin specifically improves the inefficient electron transfer between complex I and ubiquinone [181]. Furthermore, cardiolipin can be redistributed to the outer mitochondrial membrane to serve as a signal for dysfunctional mitochondria to undergo mitophagy [185]. Thus, cardiolipin is a lipid that serves as a link between mitochondrial function and the endo-lysosomal pathway, two mechanisms involved in PD and hence, providing evidence for lipids to connect both mechanisms in PD. Furthermore, several PD-related proteins are involved in lipid metabolism, including GCase.

### 6.1. Glucorerebrosidase

*GBA* encodes Glucocerebrosidase (GCase), a lysosomal enzyme that is key in the sphingolipid homeostasis by hydrolyzing glucosylceramide to form ceramide [186,187]. Homozygous mutations in *GBA* cause the lysosomal storage disorder Gaucher’s disease [188] and heterozygously mutated alleles are the most common risk factor in PD [189,190]. *D. melanogaster* Gba1a is predominantly expressed in the adult fly gut and Gba1b is expressed in the adult brain and adult fat body [191]. Loss of Gba1b results in locomotion defects and neurodegeneration [39]. Furthermore, changes in lipid metabolism and accumulation of glucosylceramide, its substrate, are observed [192]. Interestingly, numerous studies show GCase to interact with alpha-synuclein [189,193]. However, the deleterious effect of loss of Gba in alpha-synuclein expressing flies remains unclear [39,194]. Nonetheless, expression of the human PD-related-mutant *GBA* in flies induces parkinsonian phenotypes, including age-dependent locomotion defects, dopaminergic neurodegeneration and alpha-synuclein aggregation [38,40,195]. Furthermore, co-expression of mutant human *GBA* and alpha-synuclein worsens these symptoms compared to the expression of *GBA* or alpha-synuclein alone [40] providing further proof for mutant *GBA* to serve as a risk factor and for the importance of sphingolipids in PD pathogenesis.

### 6.2. Identification of Underlying Mechanisms in Non-Motor Symptoms in PD

The origin of the non-motor symptoms is poorly understood, and non-motor symptoms that pose a strong burden on the quality of life are dementia and sleep problems with the accompanying deregulation of the circadian rhythm [5,7,8]. Dementia or memory and learning difficulties have been observed in Pink1- and Parkin-deficient flies [196] and in alpha-synuclein overexpressing flies [197]; however, mechanistic insights in the development of these phenotypes have yet to be discovered. Furthermore, aged flies overexpressing the human alpha-synuclein A30P develop fragmentation of sleep that could be improved by treatment with sleep-promoting substances such as Kamikihito and Unkei-to [198]. In addition to identifying potential novel drugs to relieve sleep defects, studies in flies have also contributed to understanding the underlying mechanisms that cause these sleep defects. Using a combination of genetic tools available in the fruit fly, researchers uncovered the mechanisms underlying the deregulation of the circadian rhythm in a *pink1*- and *parkin*-mutant fly model. They identified increased ER-mitochondrial contact sites that were previously linked to PD [199], resulting in abnormal lipid trafficking followed by the depletion of the lipid phosphatidylserine at the ER. The lack of phosphatidylserine disrupts the ER-lipid balance at these ER-mitochondrial contact sites followed by the prevention of vesicle formation in neuropeptidergic neurons [200]. Interestingly, the increase of these ER-mitochondrial contact sites is sufficient to provoke sleep defects in flies [200] and the supplementation of phosphatidylserine improved these sleep defects in the PD fly models. These findings were confirmed in patient-derived fibroblasts [200], hinting towards the use of phosphatidylserine as a therapeutic agent in the alleviation of sleep problems in PD patients.

## 7. Validation in a Mammalian System

As discussed above, the use of *D. melanogaster* is valuable in the research of the underlying mechanisms in PD. One such study, identifying vitamin K2 as a modifier of the *pink1*-mutant phenotypes that is currently being tested as a therapeutic strategy in PD patients carrying *PINK1* mutations [99]. Nonetheless, research in fly models also raises caveats that one needs to be aware of. The short life span of flies (~80 days) can be used as an advantage; however, in an age-dependent disorder such as PD, this life span may be too short to observe signs and symptoms that only arise at later stages of the disease that are not represented in flies. Furthermore, while flies do contain complex neuronal connections, these connections are not per se evolutionarily conserved to the human brain. Hence, biological processes that have only been developed and evolved in a vertebrate system will not be identified in the invertebrate fruit fly. One such example is the recent finding that the mitochondrial phenotypes following loss of Parkin or PINK1 activate the STING-dependent inflammatory response. Consequently, ablation of STING prevented the inflammatory response and the accompanying neurodegenerative phenotypes [201]. *D. melanogaster* carries a fly ortholog of STING (Sting) [202]; however, loss of *Drosophila* Sting fails to suppress the behavioral and mitochondrial defects in *pink1*- and *parkin*-mutant flies [203]. Hence, flies can be successfully used for general mechanisms, but studies in mammalian/human systems are necessary to validate the relevance in the disease context and to test specific human functions.

## 8. Conclusions

PD research in flies has resulted in many different findings that are relevant for patients and that have led to the identification of new therapeutic targets. Studies in flies have significantly contributed to the identification of genetic interactions between PD-related proteins affecting similar pathways that finally converge on similar PD signs and symptoms. In addition, many of the PD-related proteins exert distinct, independent functions providing an explanation for the difference in clinical presentation and suggesting stratification of therapeutic strategies in combination with a therapy that targets the main, shared PD pathogenesis.

## Figures and Tables

**Figure 1 cells-10-00579-f001:**
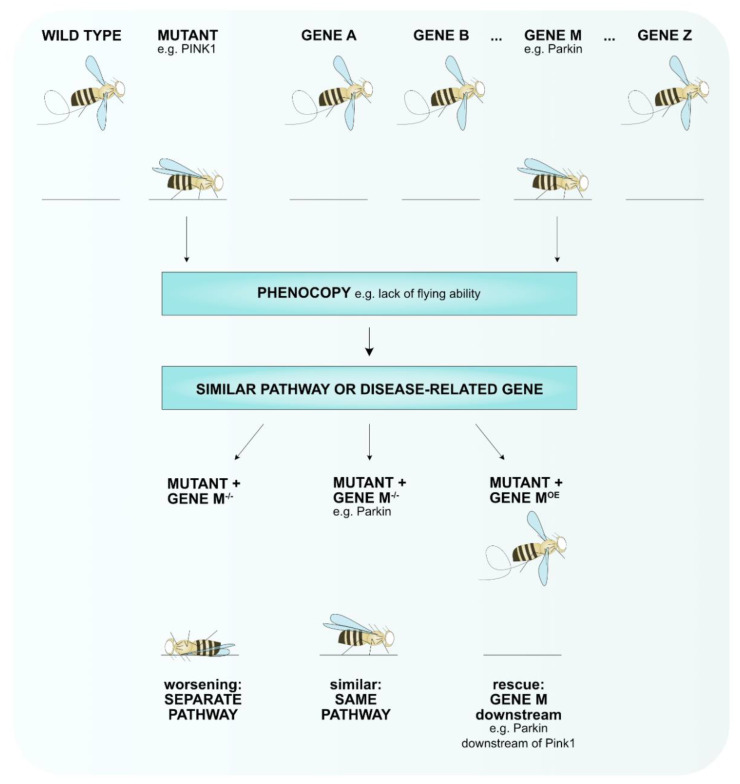
A phenotypic screen can identify novel genes involved in the disease pathway. A collection of mutant alleles (gene A–gene Z) can be used to identify genes that when mutated result in the same phenotype (phenocopy) e.g., lack of flying ability for pink1-mutant flies that are mimicked by parkin-mutant flies. To test if these genes function in the same pathway, the phenotypes of double-mutant flies can be tested. Loss of gene M worsens the phenotype of the disease mutant, thus, separate pathways play a role that converge into one. Loss of gene M does not enhance or improve the phenotype of the mutant meaning that these genes function in the same pathway (e.g., Pink1-Parkin). Overexpression (OE) of gene M rescues the mutant phenotype hints towards gene M functioning downstream of the mutant (e.g., Parkin functions downstream of Pink1).

**Figure 2 cells-10-00579-f002:**
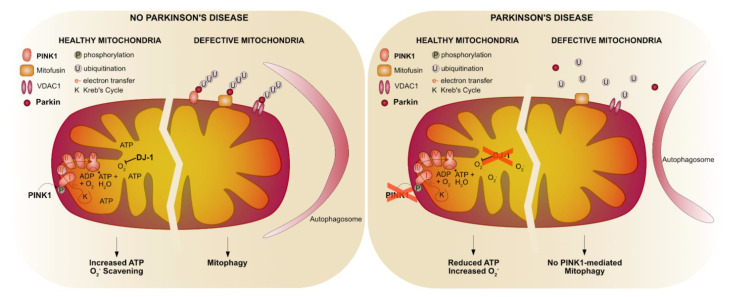
PD-related proteins in mitochondrial function. In healthy mitochondria, complex I of the electron transport chain (ETC) is phosphorylated in a Pink1-dependent fashion resulting in an efficient electron transfer through the different ETC complexes to finally use oxygen and ADP to form ATP. Oxidative stress (O_2_^.^) created by the ETC is scavenged by DJ-1. Upon a defective mitochondrial membrane potential Pink1 remains at the outer mitochondrial membrane that provides a signal for the translocation of Parkin to the mitochondria to ubiquitinate mitochondrial proteins to label these defective mitochondria for autophagosomal degradation. In genetic forms of PD, a lack of PINK1 prevents the phosphorylation of complex I that leads to an inefficient electron transfer and thus a decrease in ATP production. Furthermore, due to loss of PINK1, Parkin is not translocated to the mitochondria resulting in a lack of ubiquitination of mitochondrial proteins to label mitochondria for degradation. Loss of DJ-1 results in increased oxidative stress that results in dysfunctional mitochondria.

**Figure 3 cells-10-00579-f003:**
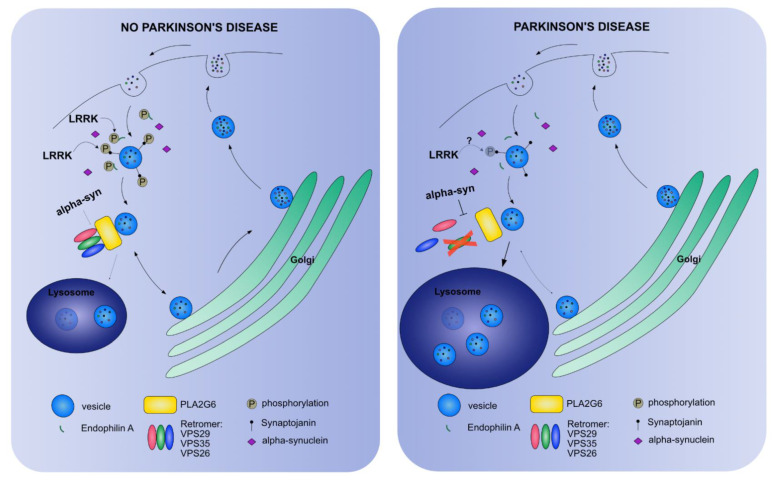
A simplified scheme highlighting processes in which PD-related proteins function in the endo-lysosomal pathway. Vesicles are recycled between the plasma membrane and the Golgi (and ER, not shown). LRRK-dependent phosphorylation of Endophilin A and Synaptojanin, and alpha-synuclein are important to facilitate in the endocytosis of vesicles. Recycling vesicles pass the retromer in conjunction with PLA2G6 to continue to the Golgi to recycle their cargo. A small number of vesicles will be transported to the lysosome to be degraded. In PD, the recycling of vesicles is disturbed by defective endocytosis or a dysfunctional retromer that is inhibited by alpha-synuclein. The effect of PD-related mutations on LRRK2-dependent phosphorylation remains unclear. The net effect is that the vesicles are not recycled to the Golgi, but are translocated to the lysosomes that expand and result in neurodegeneration.

**Table 1 cells-10-00579-t001:** Limitations and advantages of different aspects of using *D. melanogaster* as animal model to study human diseases.

Feature	Limitation	Advantage
Life cycle of ~10 days	Too short to study late-life stage signs	A lot of flies in a short amount of time
Behavior	Not all aspects can be analyzed	Locomotion, sleep, circadian rhythm, can be analyzed
Brain	Neuronal circuitry is not evolutionarily conserved	Complex neuronal circuitry (including dopaminergic neurons)
UAS-gal4 system	Off-target effects Overexpression not controlled: too much protein, and thus less physiological condition	Overexpression of human disease genes Knockdown of genes to mimic a loss of function
Genome	Only 4 chromosomes versus 23 in human	75% of the disease-causing genes have a fly ortholog

**Table 2 cells-10-00579-t002:** PD models in flies and their characteristics.

Human Gene	*Drosophila* Gene	Disease-Causing OE	LOF	KD	DA Neuron Loss	Motor Deficits	Non-Motor Symptoms	Mito Dysfunction	Endo-Lysosomal Pathway	Lipid Homeostasis	Key Findings in *Drosophila*
SNCA (AD)	/	x	/	/	+	+	+	+	+	+	- link to retromer and sphingolipids
Parkin (AR)	parkin		x	x	+/−	+	+	+			- functions in the same pathway with Pink1-circadian rhythm - age-dependent mitophagy
PINK1 (AR)	pink1		x	x	+/−	+	+	+		+	- functions in the same pathway with Parkin - circadian rhythm - age-dependent mitophagy - complex I dysfunction - lipid alterations
DJ-1 (AR)	dj-1α dj-1β		x	x		+		+			
LRRK2 (AD)	Lrrk	x	x	x	+	+			+		- link to Rab proteins-link to autophagy
VPS35 (AD)	Vps35	x	x	x	+	+			+	+	- recycling of sphingolipids
**Risk genes**											
GBA	Gba1a Gba1b	x	x	x	+	+			+	+	

/ not available; blank: not applicable; x: PD model in flies exist; + presence of phenotype; − absence of phenotype; AD: autosomal dominant; AR autosomal recessive; mito: mitochondrial; OE: overexpression; LOF: loss-of-function; KD: knockdown.

## Data Availability

No new data were created or analyzed in this study. Data sharing is not applicable to this article.

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
