# Peer review of "The Importance of Drosophila melanogaster Research to UnCover Cellular Pathways Underlying Parkinson’s Disease"

_cells, 2021, doi:10.3390/cells10030579_

Round 1

Reviewer 1 Report

Vos and Klein nicely described the importance of Drosophila melanogaster research to uncover cellular pathways underlying Parkinson’s disease. Even the authors describe many aspects of modeling the PD pathology, the authors have to introduce some aspects of dementia of PD type and maybe the aspects of 6-OHDA model. 

Author Response

We have included dementia and related observations in flies in the manuscript.

We thank the reviewer for the positive and constructive comments.

We have included 6-OHDA as drug used to model PD in the text:

‘6-hydroxydopamine (6-OHDA) is a drug that selectively destroys dopaminergic and is often used to model PD. 6-OHDA cannot pass the blood-brain barrier and therefore, has to be injected in the brain [62]. This mode of application of the drug makes it an unfavorable drug to apply in the fruit fly.’

Reviewer 2 Report

Vos and Klein present an interesting and informative review of a timely topic – that of the contribution of research in flies (Drosophila melanogaster) to the study of basic cellular mechanisms underlying Parkinson’s disease (PD). The review makes a good case for the relevance and usefulness of flies for the study of neurological diseases, PD in particular. The review focuses mainly on published data revolving around fly investigations of mitochondrial, lysosomal and lipid mechanisms and dysfunction associated with PD. The manuscript is very well written and includes insightful comments throughout. The tables and colorful figures are attractive and useful features of the article. Good consideration of caveats and limitations of PD fly studies was also provided. Overall, the present findings should be of interest to the readership of Cells and to both basic and clinical PD researchers.   

Minor suggestions/corrections to address:

1. Incorporating the following recent papers and their findings in the relevant sections of the manuscript would strengthen and ensure the timeliness of the review:

Imai, Neurosci Res, 2020: PINK1-Parkin and mitochondria

Han et al., EMBO Rep, 2020: PINK1, mitophagy

Lin et al., Mol Cell, 2020: PINK1, TUFm and mitophagy

Lin et al., Trends Endocrinol Metab, 2019:  sphingolipids

Mohite et al., ACS Chem Neurosci, 2018: alpha-synuclein mutants

Shaltouki et al., Acat Neuropathol, 2018: alpha-synuclein and mitophagy

2. The sentence toward the end of section 4.1.2 (lines 271-272) is unclear and does not make sense as written.

3. Several of the references are missing page numbers (e.g. #8, 13, 30, 176, 194).

4. There are a few relatively minor grammar issues throughout the manuscript that need attention.

Author Response

  1. Incorporating the following recent papers and their findings in the relevant sections of the manuscript would strengthen and ensure the timeliness of the review:

We apologize for not having included these articles and we have integrated most of them in the relevant sections in the manuscript. Those that were not included were left out because there are less closely related to the present flow of the manuscript.

  1. The sentence toward the end of section 4.1.2 (lines 271-272) is unclear and does not make sense as written.

We have clarified this sentence and changed this section to following text:

‘Furthermore, loss of PINK1 or Parkin results in defects in mitophagy [89] and both PINK1 and Parkin are involved in functions that are independent of one another (see 4.1.3 and 4.2). In contrast, wild type Fbxo7 that stimulates mitophagy, induces age-dependent PD-like symptoms in flies, challenging the theory that ablation of mitophagy plays a major role in PD pathogenesis.‘

  1. Several of the references are missing page numbers (e.g. #8, 13, 30, 176, 194).

We have added the page numbers to these references.

  1. There are a few relatively minor grammar issues throughout the manuscript that need attention.

We have run the Grammarly software on our text to correct grammar issues in the manuscript and have also carefully checked the manuscript for grammatical errors.

Reviewer 3 Report

In this review, the authors analyze the Parkinson Disease animal models by using the fly D. melanogaster.  Parkinson’s disease is a multifactorial disease with a varying age of onset, symptoms, and rate of progression. Its multifactorial nature determines the use of a variety of experimental models to study different aspects of the disease and to evaluate the effect of drugs in the therapies. The work is very well organized and clear. Personally, I think that it could be published in the present form. A minor suggestion, just for amore completeness, is to discuss the strengths and limitations of D. melanogaster in relation to other animal models, included mouse, rat, and non-human primate for Parkinson’s disease.

Author Response

We thank the reviewer for the positive comments and feedback. We have included a short discussion about different animal models to study PD.